# End of Green Sahara amplified mid- to late Holocene megadroughts in mainland Southeast Asia

Michael L. Griffiths [1✉], Kathleen R. Johnson [2✉], Francesco S. R. Pausata[3], Joyce C. White [4,5], Gideon M. Henderson[6], Christopher T. Wood [2], Hongying Yang[2], Vasile Ersek [7], Cyler Conrad [8,9] & Natasha Sekhon[2,10]

Between 5 and 4 thousand years ago, crippling megadroughts led to the disruption of ancient civilizations across parts of Africa and Asia, yet the extent of these climate extremes in mainland Southeast Asia (MSEA) has never been defined. This is despite archeological evidence showing a shift in human settlement patterns across the region during this period. We report evidence from stalagmite climate records indicating a major decrease of monsoon rainfall in MSEA during the mid- to late Holocene, coincident with African monsoon failure during the end of the Green Sahara. Through a set of modeling experiments, we show that reduced vegetation and increased dust loads during the Green Sahara termination shifted the Walker circulation eastward and cooled the Indian Ocean, causing a reduction in monsoon rainfall in MSEA. Our results indicate that vegetation-dust climate feedbacks from Sahara drying may have been the catalyst for societal shifts in MSEA via ocean-atmospheric teleconnections.

[1] Department of Environmental Science, William Paterson University, Wayne, NJ 07470, USA. [2] Department of Earth System Science, University of California, Irvine, CA 92697, USA. [3] Centres ESCER and GEOTOP, Department of Earth and Atmospheric Sciences, University of Quebec in Montreal (UQAM), Montreal, QC H3C 3P8, Canada. [4] Institute for Southeast Asian Archaeology, Philadelphia, PA 19104, USA. [5] Department of Anthropology, University of Pennsylvania, Philadelphia, PA 19104, USA. [6] Department of Earth Sciences, Oxford University, South Parks Road, Oxford OX1 3AN, UK. [7] Department of Geography and Environmental Sciences, Northumbria University, Newcastle upon Tyne NE1 8ST, UK. [8] Environmental Stewardship, Los Alamos National Laboratory, Los Alamos, NM 87545, USA. [9] Department of Anthropology, University of New Mexico, Albuquerque, NM 87131, USA. [10] Department of Geological Sciences, Jackson School of Geosciences, University of Texas, Austin, TX 78712, USA. ✉email: griffithsm@wpunj.edu; kathleen.johnson@uci.edu

The Southeast Asian Monsoon provides critical water resources to >600 million people each year. Even slight variations in the strength and/or timing of the monsoon can have profound societal and economic impacts on the region. Paleoclimate records have significantly advanced our understanding of the broader Asian monsoon system, particularly on orbital timescales[1]. However, due to a lack of paleoclimate records from mainland Southeast Asia (MSEA), very little is known about the range and mechanisms of Southeast Asian monsoon variability, particularly on timescales more pertinent to human occupation such as the Holocene.

The mid- to late Holocene, roughly 6 to 4 thousand years ago (ka), was characterized by one of the largest climate shifts since the last glacial termination—the end of the Green Sahara (also referred to as the African Humid Period), when a once-vegetated northern Africa transitioned to a hyper-arid desert landscape[2–5]. Both the nature and timing of this climate shift have been topics of great interest because it overlaps with societal upheavals across western Asia and the Middle East. Indeed, collapse of the Akkadian Empire of Mesopotamia[6,7], the de-urbanization of the Indus Civilization[8], and the spread of pastoralism along the Nile[9], are all examples of societal shifts that have been linked with climate extremes (e.g., the "4.2 ka event") during this period. While it has been well established that the end of the Green Sahara occurred as a result of orbital forcing amplified by vegetation/dust[10,11] and sea-surface temperature (SST)[12,13] feedbacks, the extent to which the major climate turning point of the end of the Green Sahara impacted rainfall patterns, and in turn societies, across the Southeast Asian region, has not previously been investigated.

It is particularly important to investigate Southeast Asian monsoon variability during the mid- to late Holocene transition because it overlaps with what has been termed the "missing millennia" in interior MSEA, which refers to the paucity of archeological evidence between ca. 6.0 and 4.0 ka[14,15] relative to the early and late Holocene. The Holocene prehistoric archeological record of MSEA, although still in early stages of investigation, suggests two broad periods with relatively coherent patterns: (1) an early Holocene period (ca. 9.0–6.0 ka) with mobile small societies (bands) that employed flaked stone tools, especially of river cobbles (termed "Hòabìnhian"), occupied karsts and uplands, and subsisted primarily by hunting and gathering; and (2) a late Holocene period (ca. 4.0–2.5 ka) during which nucleated settlements appear away from karst areas, including undulating lowlands, with human burials in or near the settlements, ceramics usually of elaborated styles, and at least part of their subsistence coming from domesticated plants and animals[15]. Some archeologists advocate a case of immigration of farmers (with a debated chronology) who outnumber, outcompete, and/or absorb indigenous hunter-gatherers[16–18]. However, with the exception of northern Vietnam coastal areas, there is almost no archeological evidence from interior MSEA during the millennium that immediately precedes the first appearance of societies practicing cereal cultivation. That major climate change may have been a driving factor in the societal shifts that occurred during the mid- to late Holocene in MSEA has heretofore not been considered. To this end, here we provide new insight into the potential connection between prehistoric human occupation and environmental changes in MSEA during the Holocene by comparing settlement trends in archeological data with novel paleoclimate proxy records and coupled general circulation model (GCM) sensitivity experiments incorporating a range of forcings. Our observational and model results show that weakening of the African monsoon associated with the end of the Green Sahara period amplified Holocene megadroughts in MSEA (and beyond) via cooling Indian Ocean SSTs and an eastward shift in the Walker circulation. Therefore, vegetation-dust climate feedbacks played an important role in modulating hydroclimate variability across East Asia, which may have in turn influenced human settlement patterns across the region during the Holocene.

## Results

**Multiproxy record of mainland Southeast Asian hydroclimate.** We have compiled a 9500-yr-long hydroclimate record using oxygen ($\delta^{18}O$) and carbon ($\delta^{13}C$) isotopes measured in three stalagmites (TM4, TM5 and TM11), along with radiocarbon ($^{14}C$) and Mg/Ca ratios in one stalagmite (TM5), from Tham Doun Mai cave located in northern Laos (Fig. 1; 20°45'N, 102°39'E), a region dominated by the Southeast Asian monsoon (Supplementary Fig. 1) and influenced by the El Niño/Southern Oscillation (ENSO) (Fig. 1). The stable isotope profiles for each stalagmite were constrained in absolute time with 37 $^{230}$Th-$^{234}$U ages (Supplementary Table 1), which were used to construct age models employing the Intra-Site Correlation Age Modeling (Iscam) algorithm (see "Methods" and Supplementary Fig. 2).

The composite TM $\delta^{18}O$ record, constructed by averaging the three speleothem isotope profiles for the periods of overlap, displays an increasing trend through much of the Holocene (Fig. 2a), with values increasing by ≈2‰ between the early and late Holocene. The high degree of replication between the three overlapping records (Fig. 2a, b), together with similarities with lower-resolution proxy records from China (Supplementary Fig. 3), lead us to conclude that our speleothem stable isotopes were likely deposited under equilibrium conditions and therefore reflect changes in precipitation $\delta^{18}O$ ($\delta^{18}O_p$) during the Holocene. Isotope-enabled GCM simulations and observations of $\delta^{18}O_p$ have shown that, on interannual to millennial timescales, lower $\delta^{18}O_p$ values over East Asia primarily reflects increased rainout over the Indian Ocean source region in response to increased convection and monsoonal winds upstream of the cave sites[19]. This factor is likely to dominate observed $\delta^{18}O$ signals, although other factors may also influence $\delta^{18}O_p$ over MSEA, including rainfall amount and/or shifting moisture source regions driven by various climate modes, such as ENSO[19] (Supplementary Fig. 4). We interpret the overall increase in speleothem $\delta^{18}O$ during the Holocene to reflect an overall weakening of the Southeast Asian monsoon in response to decreasing summer insolation (Fig. 2a).

The most striking feature of our record is the abrupt enrichment in $^{18}O$ beginning at ≈5.10 ± 0.07 ka (2σ), which we interpret to reflect a rapid reduction in regional monsoon intensity (Fig. 2a). This increase in $\delta^{18}O$ is coeval with an abrupt ≈5‰ two-step increase in $\delta^{13}C$ beginning at ≈5.10 ± 0.07 ka (2σ) and reaching a maximum value at 3.69 ± 0.13 ka (2σ) (Fig. 2b). Specifically, TM5 $\delta^{13}C$ exhibits a two-step transition between ca. 5 and 3.7 ka, where values increase by ~2.5‰ from ca. 5–4.6 ka, followed by a brief ≈1‰ decrease that culminates with another ~2.5‰ increase that peaks between 3.4 and 3.8 ka. The most likely explanation for this large $\delta^{13}C$ enrichment in TM5 is prior calcite precipitation (PCP) due to enhanced $^{12}CO_2$ degassing in the epikarst[20], a contention supported by the covariation with Mg/Ca ratios of the same specimen (Fig. 2c and Supplementary Fig. 5; see "Methods"). Shifts in Mg/Ca ratios in speleothems have often been tied to changes in local hydrology via PCP[20,21], where the enhanced precipitation of calcite upstream of the stalagmite during drier periods increases the Mg/Ca of the remaining cave drip water.

While it is difficult to attribute one single mechanism to changes in calcite $\delta^{13}C$, additional support for the hydrological-control on our $\delta^{13}C$ record comes from the $^{14}C$-inferred dead carbon proportion (DCP) in TM5 (Fig. 2c and Supplementary

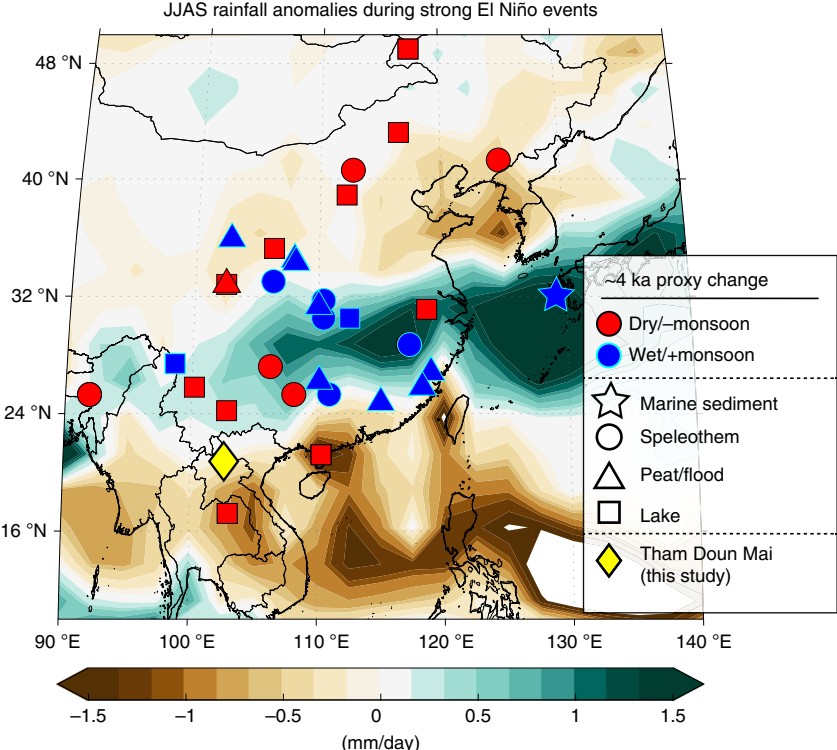

**Fig. 1 Location of Tham Doun Mai cave (diamond) and other climate proxy sites mentioned in the text.** Locations of climate proxy records across East Asia showing relative hydroclimate changes at ≈4 ka as inferred from the synthesis in Supplementary Table 1. Background shading shows Global Precipitation Climatology Project (GPCP) average June-September (JJAS) rainfall anomalies during the large 1982/1983, 1998/1999, and 2015/2016 El Niño events.

Table 3), which studies have shown to also be affected by karst hydrology[22]. The DCP in speleothems, which reflects the degree of depleted [14]C due to radioactive decay, is modulated by dissolution of the overlying [14]C-free bedrock and/or less depleted [14]C stocks in the soil and epikarst zones[23]. The relative change of DCP over time is sometimes attributed to changes in open vs. closed system dissolution, which are ultimately controlled by hydrology and the amount of void space in the epikarst, thus making DCP another effective hydrologic proxy[22]. Under this control, drier conditions result in more open-system dissolution and a lower DCP as the percolating groundwater continuously re-equilibrates with the soil $CO_2$ via air-filled voids in the epikarst. While the higher resolution $\delta^{13}C$ and Mg/Ca records show more structural features, the increasing trend in these proxies from ≈5 ka is matched by a decreasing trend in DCP approaching 4 ka, after which the trends are amplified until ≈3.7 ka. The period of lowest DCP, interpreted as drier conditions due to more open-system dissolution, is matched by the highest $\delta^{13}C$ and Mg/Ca values. These results thus support a hydrologic interpretation of both $\delta^{13}C$ and Mg/Ca variation affected by PCP and local hydrology. Worth noting, however, is that the $\delta^{13}C$ and Mg/Ca records exhibit an abrupt return to values similar to 5.5 ka by ≈3.3 ka, while the final DCP data points suggest recovery in this proxy may have been postponed up to ≈1000 years. This conflict could be a result of the disparate sampling resolutions between proxies and/or a delay in the response of the dissolution control on DCP vs. $\delta^{13}C$ and Mg/Ca controls.

Taken together, the concomitant shifts in the stable isotopes, Mg/Ca, and DCP shows that the interval from 5.11 to 3.25 ka was among the driest periods of the Holocene in northern MSEA. In this vein, two of the three speleothem records in our compilation cease [TM4: 5.31 ± 0.04 ka (2σ); TM11: 4.95 ± 0.08 ka (2σ)] and resume [TM4: 2.50 ± 0.04 ka (2σ)] growing in parallel with the

large changes in the proxies (Fig. 2), adding further weight to our interpretation. Moreover, these growth hiatuses and large enrichments in TM5 $\delta^{13}C$ also correspond with a depositional hiatus (indicating dry conditions) in nearby Lake Kumphawapi[24] located in Northeast Thailand (Supplementary Fig. 3), and are generally matched by drier conditions inferred from other regional proxy records (Fig. 1). It is worth pointing out though that there are some noticeable differences in the trends between our speleothem $\delta^{18}O$, and $\delta^{13}C$, Mg/Ca, and DCP curves, particularly between 2.5 and 6 ka. These trend differences are likely due to the fact that $\delta^{13}C$, Mg/Ca, and DCP are proxies for local water balance, while $\delta^{18}O$ is a proxy for atmospheric circulation and convective processes upstream of our cave site. For example, previously we demonstrated that the $\delta^{18}O$ "amount effect" does not dominate the modern rainfall isotope signal at our site[19], a pattern manifested in the proxy records from Tham Doun Mai[25].

**Climate change and agrarian transitions in Southeast Asia.** Archeological records from MSEA suggest that the mid- to late Holocene megadrought coincided with lifestyle changes in the region (Fig. 2d). The first appearance of cultivated cereals so far documented in MSEA, based on [14]C dates from botanical remains excavated from lowland village sites in the Mekong and Chao Phraya drainage basins, was between 4.3 and 4.0 ka, with millet in central Thailand (Non Pa Wai millet macrobotanicals)[26] and rice in northeast Thailand (Ban Chiang phytoliths excavated from inside a burial pot that contained rice)[27]. It is also inter- esting to note that in the Yangtze River Basin, weakened mon- soon rainfall ca. 5.3–4.2 ka has been linked to social and technological changes, particularly the period 4.2–4.0 ka, when major settlements and cities were abandoned due to "severe cli- matic deterioration"[28]. Meanwhile, paleoclimate archives from

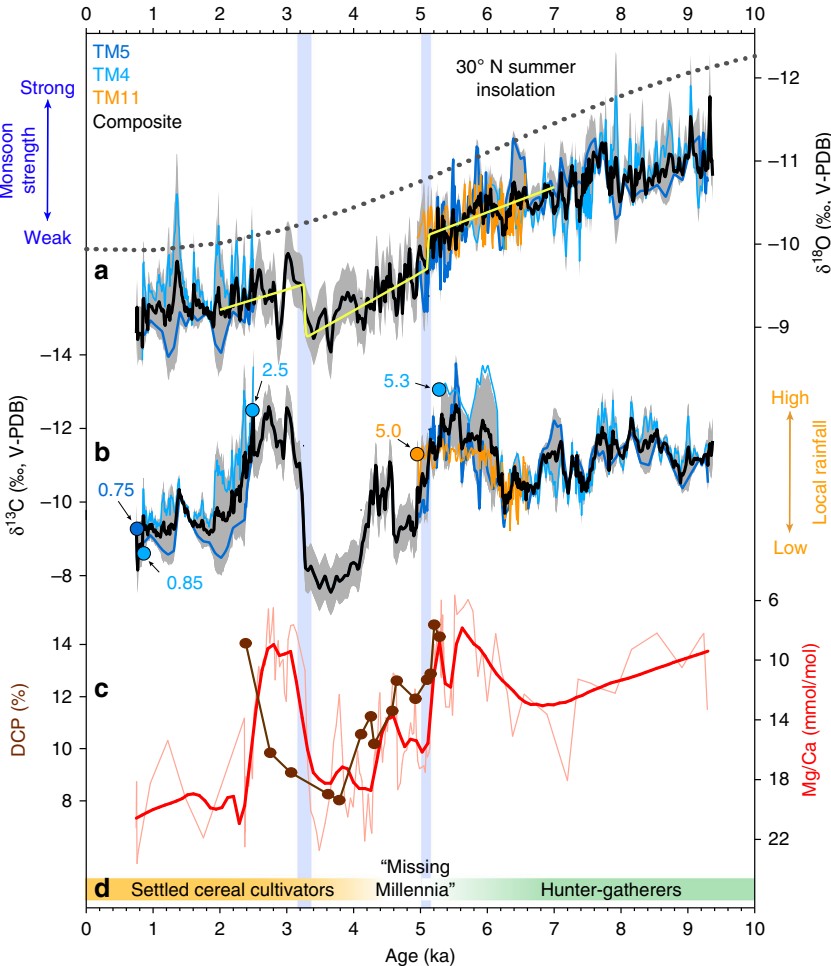

**Fig. 2 The multiproxy hydroclimate record from northern Laos and cultural shifts in mainland Southeast Asia for the Holocene. a** $\delta^{18}$O and **b** $\delta^{13}$C for stalagmites TM4 (cyan), TM5 (blue), and TM11 (orange), where values are expressed in per mill (‰) relative to Vienna Peedee Belemnite (V-PDB). Black line represents the composite record constructed by averaging data for the periods of overlap of each stalagmite record. Each record was interpolated to a common 10-year interval prior to averaging. Gray shading indicates the standard deviation for periods of overlap. For sections of the record with no overlap, the average standard deviation for overlapping periods was used. Yellow curve in panel **a** shows changepoints calculated using a Bayesian change-point algorithm that employs a probabilistic least-squares method to identify significant regime shifts[39]. Color-coded circles indicate times when the various speleothems stopped and started growing. Prior to averaging, the $\delta^{18}$O curve of TM11 was offset by −0.6‰. **c** Mg/Ca and $^{14}$C-inferred dead carbon proportion (DCP) for stalagmite TM5. **d** Approximate timing of lifestyle changes in mainland Southeast Asia during the Holocene[14].

central-eastern China indicate pluvial conditions around this time (Fig. 1 and Supplementary Table 2), concurrent with the establishment of the Xia dynasty that emerged after the so-called "Great Flood"[29,30]. Recent evidence from ancient DNA sequencing of human genomes also points to population changes in MSEA ≈ 4 ka ago[18], which the authors propose may be related to societal movements in East Asia, including some emigration into MSEA. The population movements may have been instigated in part by mid-Holocene climate changes and those movements in turn may have introduced cereal agriculture into MSEA[31]. The establishment and initial proliferation of this nucleated village agrarian lifeway in MSEA is remarkable in that it occurred during a period of extreme climate variability across the broader East Asian region. Whether or not this was a coincidence remains an open question that demands more research.

**Causes of monsoon failure during the mid- to late Holocene.**
What may have triggered such a large and abrupt megadrought in MSEA during this tumultuous time in human civilization? A common mechanism to explain this widespread drought, particularly as it pertains to the regional drying of northeastern Africa

and the Middle East around 4 ka, has been the rapid cooling of the North Atlantic[32]—the so-called "Holocene Event 3" characterized by increased ice-rafted debris. Specifically, it has been proposed that cooler North Atlantic SSTs led to deficits in Mediterranean rainfall[32], and potentially a southward shift of the Intertropical Convergence Zone (ITCZ)[33]. However, as has previously been noted (e.g., ref. [34]), the amplitude of this event was superseded by numerous other Holocene ice-rafting events (Supplementary Fig. 6), which do not coincide with large-scale megadroughts. In addition, a recent review of northern North Atlantic Holocene records showed inhomogeneous temperature trends across the region at around 4 ka, which the authors suggest is evidence that the North Atlantic did not play a critical role in the lower-latitude climate extremes around this time[35]. Thus, while we cannot completely discount the potential impact of the North Atlantic, it is unlikely that it was the primary driver of lower-latitude climate extremes at this time.

Alternatively, we hypothesize that amplifying feedbacks driven by vegetation changes and atmospheric dust loads at the termination of the Green Sahara may explain the magnitude and non-linearity of the event with respect to the gradual changes

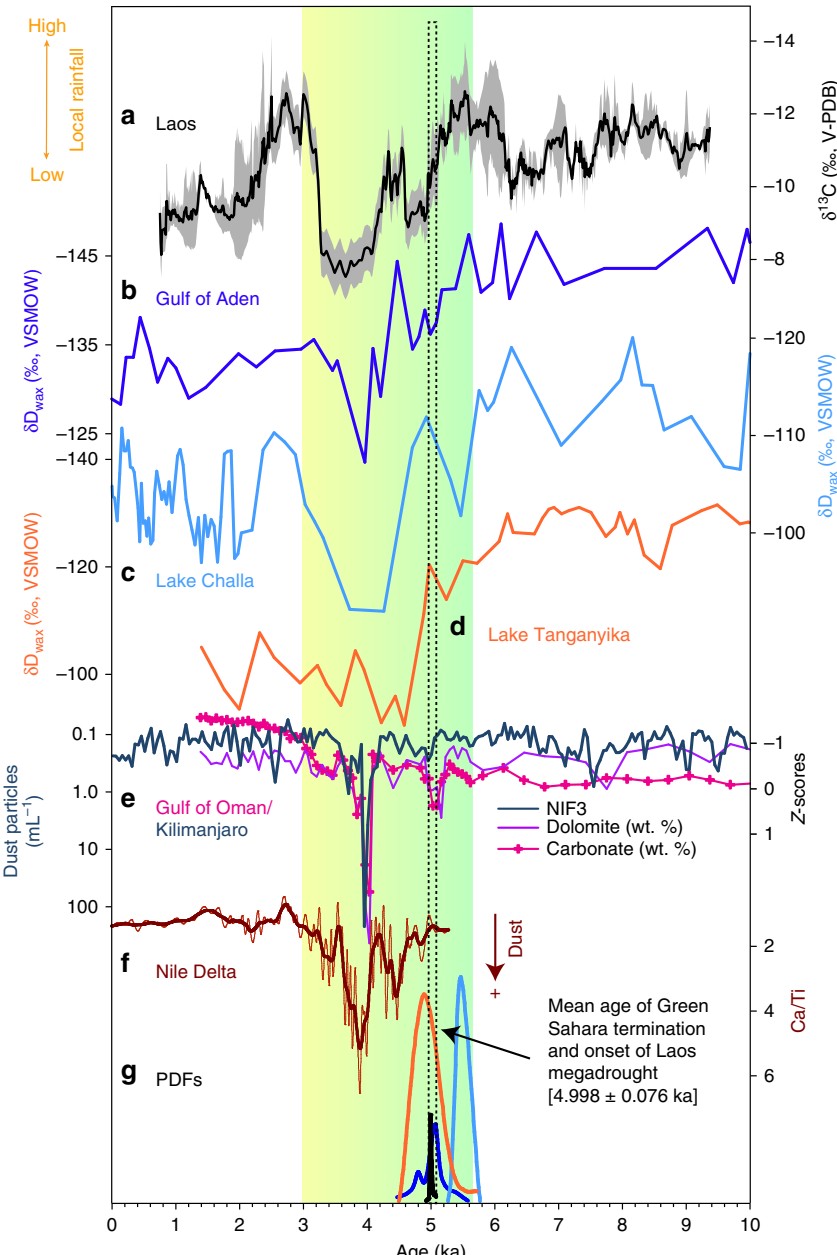

**Fig. 3 Mainland southeast Asian hydroclimate and end of the Green Sahara. a** Northern Laos composite δ¹³C record (black line) and 1σ uncertainty (gray shading) from Tham Doun Mai speleothems. **b–d** δD$_{wax}$ records from marine core P178–15P (Gulf of Aden)[3], Lake Challa[40], and Lake Tanganyika[41]. **e** Percent dolomite (light purple line) and carbonate (pink line) (expressed as standardized Z-scores) from the Gulf of Oman[33]. Also shown is the dust record (aquamarine line) from Mt. Kilimanjaro ice core KNIF3[38]. **f** Ca/Ti record of dust deposition in the Nile Delta[42]. **g** Color-coded (cyan: Lake Challa; orange: Lake Tanganyika; black: Tham Doun Mai; blue: Gulf of Aden) probability density function (PDF) output from the Bayesian change-point algorithm[39]. Vertical color bar indicates the transition from a wet to a dry Sahara between 5.5 and 3.5 ka[2,3].

in orbital forcing. The onset of severe drought conditions in MSEA at 5.10 ± 0.07 ka (2σ) coincides with the abrupt increase in dust emissions from the Sahara[2,3,36,37]. Specifically, the waning of east (Fig. 3b–d) and west (Supplementary Fig. 7) African rainfall inferred from the hydrogen isotopic composition of leaf waxes (δD$_{wax}$), and large increases in regional dust emissions centered around ≈5.0 ka and ≈4.1 ka (Fig. 3e)[7,33,36,38], generally corresponds with two episodes of reduced rainfall in northern Laos (Fig. 3a), and regional drying in the Mediterranean[34] and Middle East[7].

To quantify the timing of these climate transitions in the North/East African and MSEA records during the middle

Holocene, we applied a Bayesian change-point algorithm that uses a probabilistic, least-squares approach to identify shifts in climate regime[39]. Using the probability density function output from the statistical model (Fig. 3), which provides the age and uncertainty (2σ) on the position of the regime shifts, we performed a reduced chi-square test to detect the statistical coherence between the timing of monsoon failure across East Africa (i.e., Lake Tanganyika, Lake Challa, and the Gulf of Aden)[3,40,41] and MSEA (i.e., Tham Doun Mai). Results show that the end of the Green Sahara and onset of megadrought conditions in northern Laos was likely synchronous (χ² = 0.44; P = 0.73) with an error-weighted mean age of 4998 ± 76 (2σ) (Fig. 3); this value is in

close alignment with prior estimates from both East [4960 ± 70 (2σ)][3] and West [4900 ± 400 (2σ)][36] African proxy records (Supplementary Fig. 7). Similarly, the timing of peak megadrought conditions in northern Laos (based on TM5 $\delta^{13}C$) and maximum Saharan dust loads (Fig. 3e, f)[33,38,42] during the mid- to late Holocene, were also likely synchronous ($\chi^2 = 1.54$; $P = 0.20$).

It should be stressed, however, that factors other than Saharan dust and vegetation may have also influenced rainfall in MSEA during the mid- to late Holocene. For example, the observed return to wetter conditions in MSEA between ≈3 and 2.2 ka occurs despite the Sahara remaining generally dry. We note that similar patterns are observed in the Western Pacific SSTs[43] (Fig. 4b), whereby the mid- to late Holocene decreasing trend was interrupted by a brief, but notable, increase at ~2.5 ka, suggesting a possible shift to La Niña-like conditions. This brief hiatus in the overall drying trend is also apparent in the Lake Challa[41], and to a lesser extent the Gulf of Aden[3], $\delta D_{wax}$ records (Fig. 3b, c), further suggesting that the momentary rebound to wetter conditions was not restricted to MSEA. Furthermore, the dust records[33,38,42] (Fig. 3e, f) show that atmospheric dust loads were significantly higher between 4.2 and 3.5 ka compared with any other time during the Holocene, despite East Africa exhibiting a continued drying trend. While we can only speculate as to the driver(s) of this return to wetter conditions in MSEA around 3 ka, we hypothesize that the sudden abatement in dust loads after ≈3.5 ka meant that other forcings and feedbacks (e.g., internal ENSO variability, Northern Hemisphere summer insolation, Atlantic Meridional Overturning Circulation) became more dominant.

**Simulated dynamics of Asian monsoon hydroclimate.** To investigate the link between the desertification of a once-vegetated Sahara and monsoon failure in MSEA, we examined a series of idealized climate model simulations where prescribed Saharan vegetation and dust concentrations were altered in a way that allowed us to investigate the ocean-atmosphere feedbacks and teleconnections associated with an abrupt shift in these boundary conditions during the mid-Holocene (MH, 6 ka). Specifically, we utilized the fully coupled GCM simulations (EC-Earth version 3.1)[44] of Pausata et al.[45] and Gaetani et al.[46] to compare and contrast two MH scenarios: (1) an experiment applying MH insolation and greenhouse gases based on the Paleoclimate Modeling Intercomparison Project Phase 3/Coupled Model Intercomparison Project Phase 5 (PMIP3/CMIP5) protocol, which employs preindustrial vegetation cover and dust concentrations ($MH_{PMIP}$); and (2) an experiment in which Saharan land cover is set to shrub, and dust concentrations are reduced by up to 80% ($MH_{GS+RD}$). For both experiments, we examined 100 years (see "Methods" for details). While a transient simulation with interactive vegetation and dust emissions may be more appropriate, climate models still struggle to properly capture the abrupt transitions that occurred at the end of the Green Sahara period, with some models showing a smooth and others a more rapid ending of the Green Sahara[47]. On the other hand, the idealized nature of the simulations adopted here does allow to pinpoint the regional effect and large-scale teleconnections related to dust and vegetation changes in North Africa[48,49].

To effectively examine the climate sensitivity of Saharan dust and vegetation changes we compared the results from these two MH experiments (i.e., $MH_{PMIP}$ minus $MH_{GS+RD}$), which we refer to $\Delta MH_{PMIP}$ (Fig. 5). The model results are generally consistent with the proxies, showing widespread drying across Eurasia under reduced Saharan vegetation and increased dust emissions (Fig. 5a). In particular, the largest extreme droughts during summer (JJAS) occur in northern Africa, the Arabian Peninsula,

northern China, and the northern portions of both India and MSEA. By contrast, areas south of ≈15°N in west Africa, along with southern India and southern MSEA, exhibit increased monsoon rainfall. Together, these patterns demonstrate a significant redistribution of moisture equatorward in response to rapid land cover changes and dust emissions over the Sahara.

**Discussion**

Recent modeling experiments have demonstrated that the strengthening of the West African Monsoon (WAM) and the consequent Sahara "greening" played a dominant role in suppressing ENSO mean state and variability during the mid-Holocene[49], which might explain shifts in tropical hydroclimate through the drying of the Sahara at the end of the Green Sahara. While the ENSO phases peak in boreal winter, ENSO mode is most sensitive to perturbations applied in boreal summer from May through August[50]. Specifically, Pausata et al.[49] demonstrated that a strengthened WAM led to warm SST anomalies and a reduction in SST variability over the equatorial Atlantic. This in turn caused the Walker circulation to shift westward, which can effectively influence ENSO activity and phases through changes in the strength of the trade winds in the equatorial Pacific. As shown in studies focusing on past[49] and modern climate[51], a westward shift of the Walker circulation causes an anomalous divergent flow during summer in the central-eastern Pacific, strengthening easterly winds over the western equatorial Pacific, while weakening them over the eastern side. The weaker trades reduce the upwelling and deepen the thermocline in summer over the eastern Pacific[49], reducing the atmosphere-ocean coupling and hence decreasing ENSO variability in agreement with modeling studies[52,53]. On the other hand, the stronger trades in the central-western part of the basin causes a shoaling of the thermocline in the central Pacific during summer, leading to negative ocean temperature anomalies that travel eastward (Kelvin wave), reaching the eastern Pacific in boreal winter, and ultimately favoring the development of La Niña conditions[49,51].

We invoke a similar mechanism—albeit in the opposing direction (i.e., WAM weakening)—to explain the abrupt hydroclimate changes in MSEA observed in the proxies and model simulations. Indeed, the drying of the Sahara ($\Delta MH_{PMIP}$) (Fig. 5a) decreases the SSTs over the Indo-Pacific Warm Pool (Fig. 5b), which are characteristic of El Niño conditions. More precisely, weakening of the WAM decreases the intensity of the westerly winds along the equatorial Atlantic, which leads to increased upwelling in the eastern side of the basin (Atlantic Niña) during summer. These changes in the mean state of the Equatorial Atlantic abate and shift eastward the Walker Circulation, with a weakened descending branch (divergence) over the central Pacific eventually favoring El Niño conditions to develop (Fig. 5c). The weakened WAM can also trigger an increase in the variability of the Atlantic Equatorial mode[48], which can enhance ENSO activity as shown in previous studies (see Fig. 8 in Pausata et al.[48]). This sequence of events, thus, favors the development of cooler east Indian Ocean SSTs under the dry Sahara scenario (Fig. 5b), which reduces convection over the Southeast Asian monsoon moisture source region.

These climate model simulations are supported by the paleoclimate archives, which show evidence for overall cooler east Indian Ocean and western Pacific SSTs between 5 and 4 ka[54] (Fig. 4b, c), and conversely, trends toward higher ENSO variance (Fig. 4d–f)[55–57]. The synchronous shift to heavier $\delta^{18}O$ at Tham Doun Mai between 5 and 4 ka (Fig. 2a) is also consistent with reduced AM intensity, and more locally sourced summer monsoon moisture and/or decreased upstream rainout from the Bay of Bengal, which are all typical for El Niño events (Supplementary

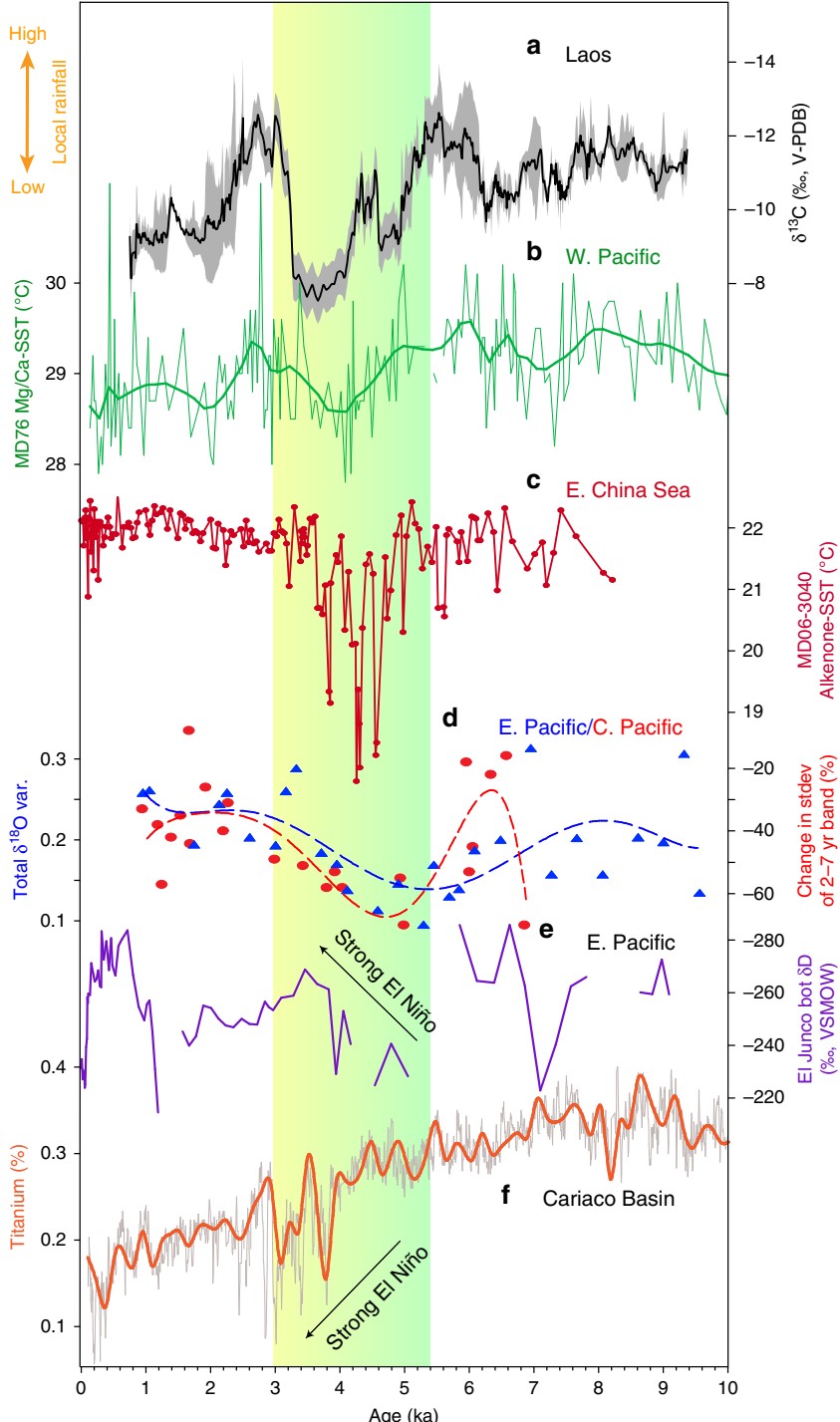

**Fig. 4 Mainland southeast Asian hydroclimate and ENSO variability. a** Northern Laos composite δ13C record (black line) and 1σ uncertainty (gray shading) from Tham Doun Mai speleothems. **b**, **c** Mg/Ca- and alkenone-inferred sea-surface temperature (SST) records from western Pacific marine cores MD76[43] and MD06–3040[73], respectively. **d** δ18O variance (var.) of individual *G. ruber* planktonic foraminifera from core V21-30 (blue triangles; eastern Pacific)[74] and relative ENSO variance changes inferred from fossil coral δ18O [calculated from sliding 30-yr windows of the standard deviation (stdev) of the 2- to 7-year band, and plotted as percent (%) differences from 1968–1998 C.E. intervals of modern coral δ18O] in Fanning Island and Christmas Island (red circles) located in the central Pacific[55]. Dashed lines show 6th order polynomial regression. **e** El Junco (Galapagos) δD of botryococcenes[56] (bot), interpreted to reflect shifts in ENSO variance. **f** Bulk titanium content of marine sediments from ODP site 1002[75] where lower values indicate drier conditions typical of El Niño events. Vertical color bar indicates the transition from a wet to a dry Sahara between 5.5 and 3.5 ka[2,3].

Fig. 4)[19]. What is more, modern climate dynamics suggest that the Asian monsoon onset is generally delayed during El Niño events due to an equatorward contraction of the ITCZ[58], a result that is manifested in the dry Sahara vs. wet Sahara model

experiments and paleoclimate records (Fig. 6). Notably, the rainfall response in MSEA to a delayed northward migration of the westerlies in the ΔMH_PMIP scenario is similar to that in northern China (i.e., drier conditions), but antiphased with

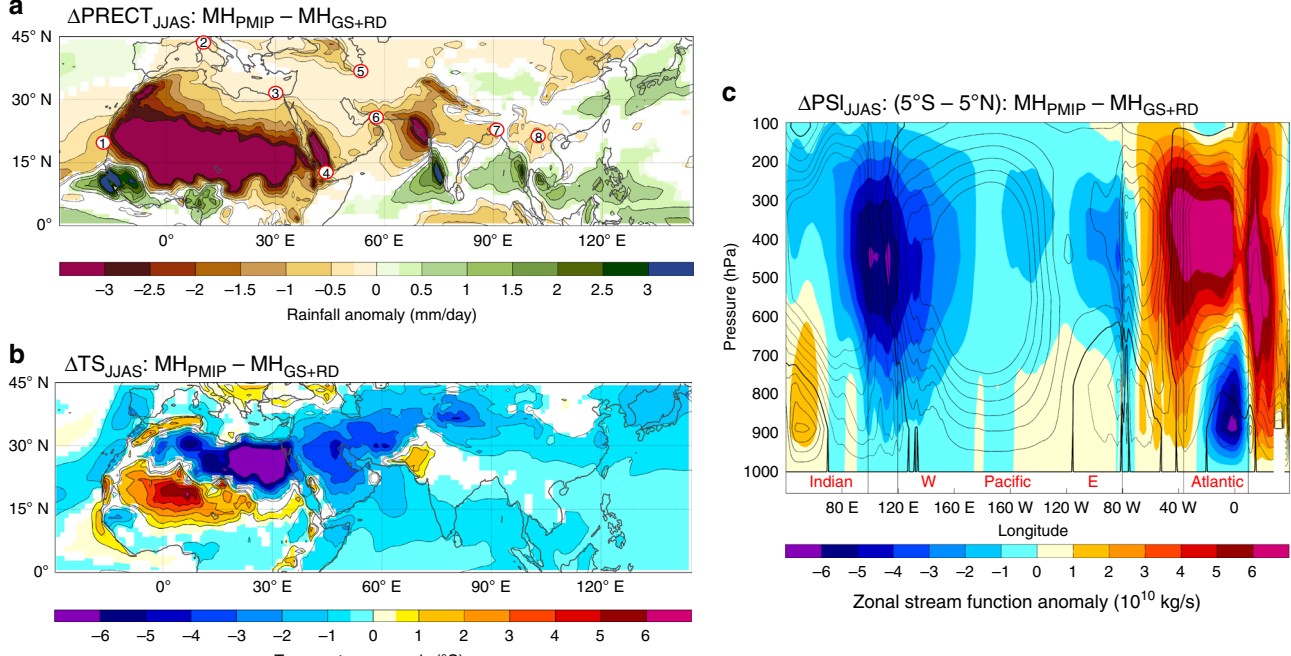

**Fig. 5 Changes in precipitation, temperature, and Walker circulation between dry and wet Sahara. a** Changes in precipitation, **b** surface temperature, and **c** zonal stream function of the Walker circulation (contour lines indicate PI climatology with a contour interval of $2 \times 10^{10}$ Kgs$^{-1}$ from $-14$ to $14$ Kgs$^{-1}$; 0 line in bold) for JJAS in the MH$_{PMIP}$ simulation relative to MH$_{GS+RD}$. Shaded regions indicate significant values at the 95% level using a two-sided $t$-test. Red circles in panel **a** denote published proxy records showing a drying trend between 5 and 4 ka: 1, West African Margin[2]; 2, Buca della Ranella[34]; 3, Nile River Delta[42]; 4, Gulf of Aden[3]; 5, Gol-e Zard[7]; 6, Gulf of Oman[33]; 7, Mawmluh Cave[8]; 8, Tham Doun Mai (this study).

central-eastern China (i.e., wetter conditions) (Figs. 5a and 6), a result that bears some resemblance to the observed modern rainfall anomalies during strong El Niño events (Fig. 1).

The simulated antiphased character of rainfall across East Asia can likely be tied to a shift in the instraseasonal stages of East Asian summer monsoon (EASM) evolution akin to the dominant "tripole" pattern of modern interannual rainfall variability[59,60]. Specifically, recent studies of both modern[60] and past[59] EASM variability hypothesize that rainfall changes over East Asia can occur from shifts in the timing and duration of the EASM instraseasonal stages (i.e., spring, pre-Meiyu, Meiyu, and mid-summer), which are ultimately linked with the south-north displacement of the westerlies relative to the Tibetan Plateau. Recent work by Zhang et al.[59] showed that a seasonal delay in the northward migrating westerlies during spring/summer—similar to the results of this study (Fig. 6a)—is what led to overall higher rainfall anomalies in central-eastern China during the deglacial cooling events (i.e., Heinrich Stadial 1 and the Younger Dryas), which they showed was due to a lengthened Meiyu and shortened midsummer stage. We propose a similar mechanism here for the antiphase behavior of EASM rainfall under the $\Delta$MH$_{PMIP}$ scenario. Dust-source paleoclimate records from the Japan Sea[61] support our model simulations, showing that the westerlies were indeed shifted southward between 4 and 5 ka (Fig. 6c), resulting in extreme drought conditions in Northern China and MSEA, coincident with an equatorward contraction of the ITCZ (Fig. 6c).

To conclude, we have provided the first evidence for a link between the termination of the Green Sahara and widespread declines in monsoon rainfall across interior MSEA. This work highlights the sensitivity of Southeast Asian hydroclimate to large and abrupt shifts in Earth's boundary conditions, and in particular, demonstrates the potential for densely populated regions of East Asia to rapidly switch between wet and dry background climate states. The long-term and sub-regional societal responses

to these profound and at times abrupt climate shifts remain to be elucidated through archeological investigations.

## Methods

**Cave location and speleothem samples.** Tham Doun Mai Cave is located in northeastern Laos (20°45′N, 102°39′E, 360 m a.s.l.) adjacent to the Nam Ou River, close to the border of Vietnam. This ~3745 m long cave is extremely well-suited for paleoclimate reconstruction as it is hydrologically active, has a stable temperature and high relative humidity (Ave. Temp. = 22 ± 0.38 °C; RH = > 95%), contains numerous actively forming stalagmites, and has only one known small entrance. The three speleothems used in this study (TM4, TM5, and TM11) were collected from Tham Doun Mai in 2010 (Supplementary Fig. 1). TM4 and TM5 were collected from the cave passage ~150 m from the entrance, while TM11 was collected from an upper chamber of the cave located slightly closer to the cave entrance. X-ray diffraction reveals that all specimens are composed of 100% calcite. Prior to analysis, each stalagmite was sectioned in half along the growth axis, and later polished to help identify the central growth axis.

**Stable isotopes.** Samples for isotope ratio measurements were drilled along the stalagmite's central growth axis. Stalagmite surfaces and drill bit were cleaned with ethanol prior to sampling. The older portion (pre-hiatus, 5.3 to 9.4 ka) of TM4 was sampled at 250 μm resolution (~10 years) using a Sherline micromill and the younger portion was sampled at an average resolution of 76 μm (~3 years) using a New Wave MicroMill drill at the University of California Irvine (UCI). The younger portion (0.75 to 5.30 ka) of TM5 was sampled at 500 μm resolution (~17 years), while the older portion (5.30 to 9.30 ka) was sampled at 1 mm resolution (~38 years). Stalagmite TM11 was sampled at 500 μm resolution (~9 years).

Powdered calcite samples (~30–70 mg) were analyzed for stable isotope composition utilizing a Kiel IV carbonate device coupled with a Thermofinnigan Delta V Plus isotope ratio mass spectrometer at UCI. A total of 16 standards (NBS-19, NBS-18, and OX, an in-house quality control standard) were analyzed during each run of 30 unknown samples. Results of the isotopic analysis are expressed in per mill (‰) relative to Vienna Pee Dee Belemnite (V-PDB) standard using the delta notation, defined as: $\delta^{18}O = [(^{18}O/^{16}O)_{sample}/(^{18}O/^{16}O)_{standard} - 1] * 1000$. The standard deviation (i.e., analytical precision) of repeated NBS-19 measurements is 0.06‰ for $\delta^{18}O$ and 0.03‰ for $\delta^{13}C$. A total of 2152 oxygen and carbon isotope measurements were conducted, comprising 1022, 388, and 742 analyses for stalagmites TM4, TM5, and TM11, respectively. However, large age reversals in the U-series dates of TM11 for the middle and lower portion of the stalagmite (10.31–6.68 ka), which we believe is due to the presence of microsparite

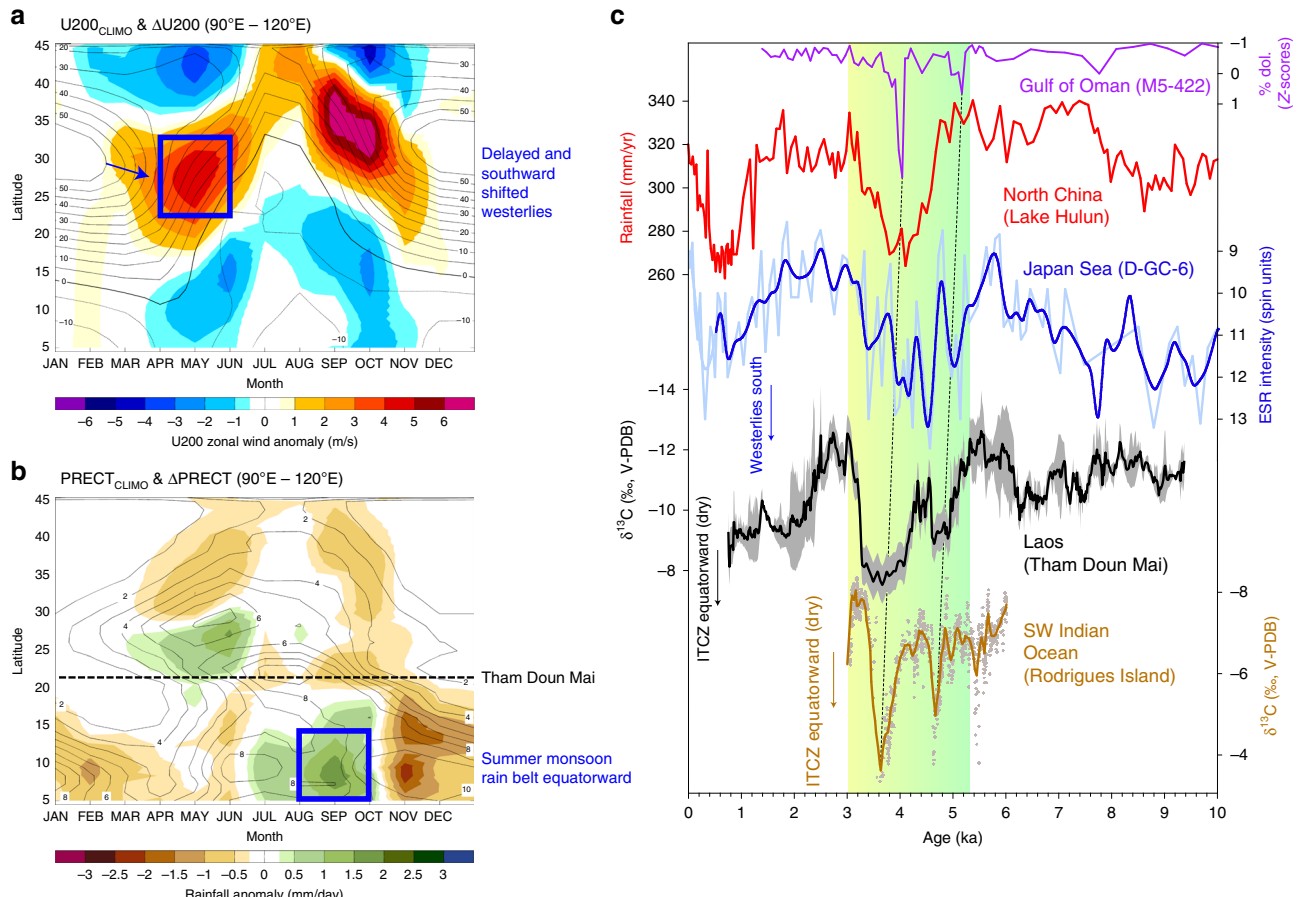

**Fig. 6 Seasonal shifts in the westerlies and precipitation across East Asia between dry and wet Sahara. a, b** Hovmöller diagrams of climatological U200 winds and precipitation for wet Sahara and its anomalies (contours: $MH_{GS+RD}$ climatology; shading: difference between $MH_{PMIP}$ and $MH_{GS+RD}$ experiments). **c** Support for the model simulations is provided by the paleoclimate archives, which show drier conditions in North China (red)[76] due to a southward shift in the westerlies (blue; inferred from ESR intensity of silt-sized quartz grains in sediments from the Japan Sea)[61] and equatorward contraction of the ITCZ during boreal summer (black; this study) and austral summer (brown)[77]. Vertical color bar indicates the transition from a wet to a dry Sahara between 5.5 and 3.5 ka[2,3].

which could indicate diagenesis, precluded us from constructing a reliable age model for the lower-middle sections of the record. Hence, only 191 analyses were included in the composite record. Omission of these 551 isotope values from the composite record has no impact on the overall conclusions of the paper given that the focus is on the 5–4 ka period. Moreover, the most interesting aspect of the TM11 record is the stoppage in growth at around the same time as TM4, which is coincident with the large $\delta^{13}C$ and $\delta^{18}O$ enrichments of TM5, interpreted to reflect a large reduction in cave recharge and monsoon weakening respectively.

**Trace elements (Mg/Ca).** Mg/Ca was analyzed in TM5 calcite powders using splits from the material originally drilled for stable isotope analyses. Samples were run on a Nu Instruments Attom High-Resolution Inductively Coupled Plasma Mass Spectrometer (HR-ICP-MS) at the Center for Isotope Tracers in Earth Science (CITIES) laboratory at the University of California, Irvine. Raw intensities were converted to concentrations utilizing an external calibration curve constructed via analysis of five standards of known concentration and a blank. All samples, standards, and blanks were spiked with an internal Sc-Ge standard prior to analysis to allow for instrument drift correction. One-hundred eighty-four total samples were analyzed for Mg and Ca with an average uncertainty in the Mg/Ca ratio of 3% (1σ). Samples were run at higher resolution during the period of interest (5.5 to 2.5 ka). $\delta^{13}C$ and Mg/Ca in TM5 covary ($R^2 = 0.55$) across the time series (Supplementary Fig. 5), suggesting a similar control on both proxies in this sample. Increases in $\delta^{13}C$ (less negative) coincide with increases in the Mg/Ca ratio, behavior that matches the potential effect of PCP. Therefore, the Mg/Ca data provides additional evidence for a hydrologic interpretation of $\delta^{13}C$ variability in TM5. An unmatched negative excursion in Mg/Ca during the long-term increase in both proxies around 4 ka may suggest a separate control or shorter response time active for Mg/Ca during this dry period.

**Radiocarbon ($^{14}C$).** Fifteen samples from TM5 were analyzed for $^{14}C$ at the W.M. Keck Carbon Cycle Accelerator Mass Spectrometer at the University of California, Irvine, and the resulting data was used in conjunction with previously acquired U–Th dates to determine the DCP (Supplementary Table 3). Small pieces of calcite were extracted from the central growth laminae of TM5 using a rotary Dremel drill. Carbonate subsamples were leached in 10% HCL, and then hydrolyzed in 85% $H_3PO_4$. Following conversion to $CO_2$, samples were graphitized via iron catalyzed hydrogen reduction[62]. $^{14}C$ measurements were made on an NEC Compact (1.5 SDH) AMS system, using six aliquots of Oxalic Acid I as the normalizing standard. DCP was calculated following methods described by ref. [63], utilizing IntCal13 data for the atmospheric $^{14}C$ activity at the time of formation[64]. The resulting time series is compared with the $\delta^{13}C$ and Mg/Ca data in Fig. 2, which shows similar trends in all three proxies from ≈5–3.5 ka. Specifically, the increase in $\delta^{13}C$ and Mg/Ca from ≈5 ka is matched by a decreasing trend in the DCP record. The period of lowest DCP, interpreted as drier conditions due to more open-system dissolution, is matched by the highest values of $\delta^{18}O$, $\delta^{13}C$, and Mg/Ca. These results support a hydrologic interpretation of $\delta^{13}C$ variation affected by PCP via shifts in local hydrology.

**$^{230}Th$-$^{234}U$ dating and age models.** Thirty-nine subsamples for dating were obtained by cutting out solid chunks parallel to speleothem growth bands with a Dremel rotary tool with a diamond bur. For U–Th analysis, 0.1–0.2 g samples were extracted every 1–2 cm along the growth axes for each stalagmite, including above and below any suspected hiatuses. Samples were cleaned in an ultrasonic bath with isopropanol and Milli-Q water, dried in oven on foil, and stored samples in clean 1.5 mL Eppendorf microcentrifuge tubes. The U–Th dating was conducted at the University of Oxford on a Nu instruments multi-collector inductively coupled plasma mass spectrometer (MC-ICP-MS). Calcite samples were dissolved, spiked with a mixed $^{229}Th$-$^{236}U$ spike, and purified by ion-exchange chemistry; these

procedures for chemical separation and purification are similar to those adopted in ref. [65], and the chemical and mass spectrometric approaches broadly follow the techniques described in refs. [66,67]. U–Th ages were corrected for the presence of small amounts of initial Th utilizing the average crustal ($^{230}$Th/$^{232}$Th) value of 1.21. The uncertainties corresponding to this initial Th correction are arbitrarily assigned to be 50% of the ($^{230}$Th/$^{232}$Th) ratio.

The age models for all depth-isotope series were calculated using Iscam[68] "Intra-Site Correlation Age Modeling"). This method calculates a point-wise linear interpolation between adjacent dates based on the highest correlation obtained between multiple δ$^{13}$C or δ$^{18}$O time series (within their age uncertainties) calculated from 100,000 Monte-Carlo (MC) simulations; given that our δ$^{13}$C records exhibit more variability than the δ$^{18}$O for the Holocene, we used the δ$^{13}$C records for the Iscam analysis, as there were more "tie points" to help align each series within their age uncertainties. Age model uncertainty (i.e., 68%, 95%, or 99% confidence intervals) was performed against a red-noise background using 2000 pairs of artificially simulated first-order autoregressive time series (AR1). For a more detailed description of the age model algorithm and age model uncertainty calculation, we refer the reader to ref. [68]. As the three stalagmites (TM4, TM5, and TM11) were collected from nearby locations within Tham Doun Mai, it is highly likely that the common geochemical excursions between the different records reflect climate variability. The likely reason for slight differences in the three records is that they formed under different drips, and δ$^{13}$C is controlled by PCP and therefore drip/flow rate. The fact that two of the stalagmites (TM4 and TM11) stopped growing for a period of time while the other (TM5) continued to grow, clearly suggests that they have different local hydrology, which will lead to differences in the details of their δ$^{13}$C.

**Climate model simulations.** We adopted the simulations performed in Pausata et al.[49] and Gaetani et al.[46] who used the Earth system model EC-Earth[44] version 3 to perform a set of numerical simulations of the middle Holocene. The motivation for employing such a set of simulations arises from the fact that more traditional modeling experiments, such as the PMIP MH experiments, greatly underestimate the magnitude and scale of rainfall amount during the Green Sahara. Specifically, as shown by Tierney et al.[2], it is only when dust and vegetation feedbacks are included in the model[45], can it most accurately simulate Green Sahara conditions as indicated from the proxies. Thus, in order to robustly simulate the ocean-atmospheric teleconnections, it is necessary to use the modeling framework that best captures the climate response to these forcings and feedback processes.

The atmospheric model is based on the Integrated Forecast System (IFS cycle 36r4) developed by the European Center for Medium-range Weather Forecasts, including the H-TESSEL land model. The simulations were run at T159 horizontal spectral resolution, corresponding to roughly 1.125° by 1.125° and at a vertical resolution of 62 vertical levels. The ocean model is the Ocean General Circulation Model —NEMO version 3.3.1[69]. It solves the primitive equations discretized on a curvilinear horizontal mesh with a horizontal resolution of about 1° 1° and 46 vertical levels. At the surface, the model is coupled every model hour with the Louvain-la-Neuve Ice Model—LIM3[70] having the same horizontal resolution as NEMO. EC-Earth has been extensively used for simulating past, historical and future climate contributing to the Fifth Assessment Report of the Intergovernmental Panel on Climate Change and to the Paleoclimate Modeling Intercomparison Project. EC-Earth has shown good skills in representing monsoonal precipitation both temporally and spatially in present day climate[45]. Boundary conditions for the middle Holocene experiments were set at preindustrial values according to the PMIP/CMIP5 protocol with the exception of the orbital forcing that was set at 6 ka values and computed internally using the method of Berger[71], and the greenhouse gases that follow the PMIP3/CMIP5 protocol. Vegetation cover and properties, and dust concentrations were prescribed. In the MH$_{PMIP}$ experiment, the dust climatology is based on the long-term monthly mean (1980–2015) of the MERRAero data as the Community Atmosphere Model (CAM)[72], which is used in the CMIP5 and has biased dust emissions over the Sahara region[46]; see Supplementary Fig. 1 in Gaetani et al.[46] for more details. Additional information on the MERRAero dataset can be found at https://gmao.gsfc.nasa.gov/reanalysis/merra/MERRAero/. In the MH$_{GH+RD}$ experiment, the vegetation type over the Sahara domain (11°–33° N and 15° W–35° E) is set to shrub (MHGS) and the dust amount is also reduced by up to 80% (Fig. 1 and Extended Data Fig. 1 in Pausata et al.[49], based on recent estimates of Saharan dust flux reduction during the MH[36]. The change in vegetation cover from shrub (MH$_{GS+RD}$) to desert (MH$_{PMIP}$) corresponds to an increase in the surface albedo from 0.15 to 0.30 and a decrease in the leaf area index from 2.6 to 0.2 (see Table 1 in ref. [49]). The dust changes from the MH$_{GH+RD}$ to the MH$_{PMIP}$ correspond to an increase in the global total AOD of 0.02 (Fig. 1 in ref. [49]). The changes in dust concentration and vegetation cover were not meant to provide a faithful representation of the MH conditions over the Sahara and nearby regions, but to provide insight on their potential feedbacks.

## Data availability
Data from this article can be downloaded from the cave section of the National Oceanic and Atmospheric Administration National Centers for Environmental Information Paleoclimatology archive: https://www.ncdc.noaa.gov/paleo/study/.

## Code availability
We used the Matlab code by Ruggieri (ref. [39]) for the Bayesian change-point detection in our records and those from East Africa. The composited rainfall map displayed in Fig. 1 was provided by the NOAA/OAR/ESRL PSD, Boulder, Colorado, USA, from their Web site at https://www.esrl.noaa.gov/psd/. The Matlab script to produce Fig. 1 is available from the authors upon request. We used the Matlab code of Fohlmeister (ref. [68]) to generate the U–Th age models.

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

## Acknowledgements

We thank Bounheuang Bouasisengpaseuth, Norseng Sayvongdouane, Sengphone Keophanhya and other participants in the Middle Mekong Archeological Project and Lao government officials and departments for their assistance with the fieldwork, which was funded in part by Henry Luce Foundation grant to the University of Pennsylvania Museum. We thank Russell N. Drysdale for useful discussions, and Dachun Zhang and John Southon for help with anlayses. The climate research was supported by National Science Foundation awards 1405472 and 1603056 to K.R.J., and awards 1404932 and 1602947 to M.L.G. F.S.R.P. acknowledges funding from the Swedish Research Council (FORMAS) as part of the Joint Programming Initiative on Climate and the Belmont Forum for the project "Palaeo-constraints on Monsoon Evolution and Dynamics (PACMEDY). In addition, we acknowledge support from the NSF Doctoral Dissertation Improvement Grant #1724202 awarded to C.C. This research was also supported by a 2010–2012 NOAA/UCAR Climate and Global Change Postdoctoral Fellowship to M.L.G.

## Author contributions

M.L.G. and K.R.J designed or conceived the study. F.S.R.P. analyzed and interpreted the climate model experiments. M.L.G. and K.R.J. were responsible for designing the study. F.S.R.P. carried out the climate model experiments. H.Y., G.M.H., and V.E. performed

the $^{230}$Th-$^{234}$U dating. H.Y. helped collect the speleothem samples and carried out the speleothem micromilling and stable isotope analysis in the laboratory of K.R.J., and $^{230}$Th-$^{234}$U dating in the laboratory of G.M.H. C.T.W. and K.R.J. conducted the Mg/Ca and $^{14}$C measurements. N.S. helped with micromilling the speleothem samples and stable isotope analysis in the laboratory of K.R.J. All authors contributed to writing of the manuscript.

## Competing interests
The authors declare no competing interests.
