## [Peer Review File · Nature Communications]

Reviewers' comments first round:

Reviewer #1 (Remarks to the Author):

Review of "Mid-to-Late Holocene Megadroughts in the Middle Mekong Basin Linked to Global Climate Changes" by Griffiths et al., 2020

This study investigates the relationship between the end of African Humid Period and cultural shifts in Mainland South East Asian (MSEA) as consequence of Megadroughts. The study is novel and overall well-written, however I think that further analysis is needed about the invoked mechanisms explaining the Walker circulation shift due to ENSO reinforcement during the mid-to-late Holocene period.

Major comments:

Ln 45: I found confusing using "mid" and "middle" Holocene when you refer to the transition period between 6-4 kya when you have Megadroughts, especially because the mid-Holocene is usually considered as a humid period. Consider to replace with "mid-to-late Holocene transition". Same at Ln 150 and somewhere else in the text.

Ln 84-86: I do not understand why mentioning that Tham Doun Mai cave is located in a region strongly influenced (*now-a-days*) by ENSO. I also do not understand why in Fig.1 you show precipitation anomalies (I guess from model simulation? Not specified in the caption - please provide further information) during strong ENSO events in the 20-21 century. Additionally ENSO peak occur during NDJ... why plotting JJA anomalies? Furthermore, in the rest of the study you show JJAS anomalies.

What do you want to show here? What is the connection with what you want to explain? Please consider to rethink about the figure or better address questions above. Furthermore see other questions about the relationship with ENSO - Walker relationship below "Discussion".

Ln 197: Here you refer to fig. 3 as "using the probability density function output from the statistical model...", while fig. 3 shows climate and societal changes during the Holocene in MSEA and links with the end of the Green Sahara. In the first place, the reader cannot find immediately the information you are talking about, because you refer to the small part of the plot at the very top only at the very end of the caption. I suggest to move the density functions below in the figure, after subplot "f) Nile..", naming it as "g) PDFs" or something like that, and in the text at Ln 197, next to Fig. 3, I would refer to Fig. 3g. Also please, make this part of the plot bigger...

Figure 4 and 6 in the text are not in vectorial format. Please check it.

"Discussion". I think that this section about invoked mechanisms needs some further analysis, e.g. what is the relationship between ENSO and Walker circulation during the investigated season (JJAS)? ENSO peaks occur during NDJ, while you are investigating changes in Walker circulation during JJAS. How do you explain this inter-seasonal relationship?

LN: 248 - 257 This paragraph is not clear to me. It seems that weakened WAM is causing the ENSO reinforcement. Please also refer to "Towards understanding the suppressed ENSO activity during mid-Holocene in PMIP2 and PMIP3 simulations" by Lin Chen, Weipeng Zheng, Pascale Braconnot for further discussion.

Reviewer #2 (Remarks to the Author):

There is one important point that the authors might consider, because it does reflect on how

scholars in the other disciplines might interpret the results. What was the geographical extent of the drought area in Northern MSEA at about 5000 years ago? This is important, because the paper also raises, here and there, the likelihood that other regions to the immediate south, and in parts of China, actually received more rainfall at this time (e.g., lines 234–235 in the manuscript). In other words, the whole 'megadrought' concept does need very careful delineation in terms of exactly where the megadrought is considered to have happened. For cultural matters, such as migrations of human populations, this is likely to be very important.

Reviewer #3 (Remarks to the Author):

The revised manuscript by Griffith et al. is an improvement of the previous version. The drought event occurred about 5.11~3.25 ka has been well supported by the stable isotope compositions, Mg/Ca ratio and the dead carbon proportions. By using the model simulations, it was proposed that reduced Saharan vegetation and increased dust emissions might contribute to the summer drought in northern Africa, the Arabian Peninsula, northern China, and the northern portions of both Indian and MSEA. This proposition is intriguing and deserved for studying by using both observational records and model simulations. However, there are still some issues need to be clarified before the manuscript could be published. I suggest a moderate reversion.

Firstly, with the model simulations, I could accept that the end of green Saharan and increased dust emission may shift the Walker circulation eastward and cooled the Indian Ocean, causing a reduction in monsoon rainfall in MSEA and other regions. But I am curious that why the rainfall increased again while Saharan desert was still there and dust emissions was higher after 3.25 ka. The ending of this drought is also very important for us to understand the event. It suggests that a returning mechanism might be inherent to this event, e.g. recovering of the North Atlantic thermohaline circulations, although it is unlikely to be the primary driver as suggested by the authors. The authors should explain this accordingly, as it is closely related to the development of this event, and may provide clues to disclose the mechanism of this event.

Secondly, the authors state "The establishment and initial proliferation of this nucleated village agrarian life way in MSEA is remarkable in that that it occurred during a period of extreme climate variability across the broader East Asian region." With the record reported in this study and other records mentioned, I agree to receive that the period from 5.1-3.2 ka is of extreme climate variability across the broader East Asia. However, there is no evidence or reference showing that the establishment and initial proliferation of this nucleated village agrarian life way occurred in MSEA during this period in the revised manuscript. The first appearance of cultivated cereals documented in MSEA, as mentioned in the text, is hardly to support this assessment. Although two broad periods, i.e., an early Holocene period and a late Holocene period, can be grabbed with the investigation of archeological records in this region, however, how and when the nucleated village agrarian life established and initially proliferated was still unclear. So I suggest the author to reorganize this section and demonstrate the link between the climate change and lifestyle change with more solid evidences. This is because the authors omitted the archeological evidence and discussions in the reversion.

Here, the authors also refereed the linkage between weakened monsoon rainfall in the Yangtze River Basin and social and technological changes ca. 5.3-4.2 ka to indicate the climate change and its influences on human societies during the middle Holocene. However, as shown in the following sections, the drought within the middle Holocene in MSEA was largely linked with a dry climate in Northern China but a wet climate in central Eastern China. So the weakened monsoon rainfall in the Yangtze River Basin ca. 5.3-4.2 ka would be not the case. The authors need to double check the fact and also the potential dynamics which may contradict with the fact.

The open-system dissolution could well explain the decreasing trend of DCP started from beginning of the drought event. However, at the end of this event (e.g. 3-2 ka), the DCP data show a reversed relationship with the hydro-climate condition. How to interpret the data within this interval is also very important for fully understanding this record.

Line 284-286, it was stated " the westerlies were shifted southward between 4 and 5 ka (Fig. 6c),

resulting in extreme drought conditions in Northern China and MSEA, and coincident with an equatorward contraction of the ITCZ." Maybe I misunderstand here. The hypothesis for the drought event between 4-5 ka in MSEA is that the end of green Saharan and increased emission of dust lead to increased activity of ENSO and then delayed onset of Asian summer monsoon, not the southward shift of westerlies!

Mid-to-Late Holocene Megadroughts in the Middle Mekong Basin Linked to Global Climate Changes

Michael L. Griffiths, Kathleen R. Johnson, Francesco S.R. Pausata, Joyce C. White, Gideon M. Henderson, Christopher T. Wood, Hongying Yang, Vasile Ersek, Cyler Conrad, Natasha Sekhon

Reviewer #1 (Remarks to the Author):

This study investigates the relationship between the end of African Humid Period and cultural shifts in Mainland South East Asian (MSEA) as consequence of Megadroughts. The study is novel and overall well-written, however I think that further analysis is needed about the invoked mechanisms explaining the Walker circulation shift due to ENSO reinforcement during the mid-to-late Holocene period.

Reply: We thank the referee for encouragement and constructive review.

Major comments:

Ln 45: I found confusing using “mid” and “middle” Holocene when you refer to the transition period between 6-4 kya when you have Megadroughts, especially because the midHolocene is usually considered as a humid period. Consider to replace with “mid-to-late Holocene transition”. Same at Ln 150 and somewhere else in the text.

Reply: Thank you, done.

Ln 84-86: I do not understand why mentioning that Tham Doun Mai cave is located in a region strongly influenced (*now-a-days*) by ENSO. I also do not understand why in Fig.1 you show precipitation anomalies (I guess from model simulation? Not specified in the caption - please provide further information) during strong ENSO events in the 20-21 century. Additionally ENSO peak occur during NDJ... why plotting JJA anomalies? Furthermore, in the rest of the study you show JJAS anomalies. What do you want to show here? What is the connection with what you want to explain? Please consider to rethink about the figure or better address questions above. Furthermore see other questions about the relationship with ENSO - Walker relationship below “Discussion”.

Reply: We apologize for any confusion. We have updated the caption to better conform with the data presented in Fig. 1. Namely, we have now indicated in the caption that the data are instrumental (GPCP) composite rainfall anomalies (relative to 1981-2010) for the 1982/83, 1998/99, and 2015/16 years. In addition, to be consistent with the model figures, we have now calculated composite anomalies for JJAS, rather than JJA as was displayed in the original submission.

We are aware that ENSO peaks during boreal winter, however, instrumental data show that it can also impact summer monsoon rainfall across Asia (Fig. 1), before peaking in the following winter. We chose to focus on these months because >70% of the annual rainfall at our study site falls during boreal summer, thus being the critical season for evaluating the impacts of ENSO.

To summarize, the motivation for plotting up the modern summer monsoon rainfall anomalies (Fig. 1) during strong El Niño events is to highlight the modern influence of ENSO on rainfall anomalies across MSEA, and thus add weight to the results of our model simulations (Fig. 5a).

Hence, the spatial pattern of summer monsoon variability across East Asia (due to an eastward shifted Walker circulation) in our mid-Holocene model simulations are somewhat consistent with the patterns observed in the instrumental and proxy records, providing confidence that the model is successfully representing the rainfall patterns across East Asia under enhanced “El-Nino-like” forcing. In the original submission, we attempted to make this connection:

Line 349: *“Notably, the rainfall response in MSEA to a delayed northward migration of the westerlies in the ΔMH_{PMP} scenario is similar to that in northern China (i.e. drier conditions), but antiphased with central-eastern China (i.e. wetter conditions) (Figs 5a and 6), a result that bears some resemblance to the observed modern rainfall anomalies during strong El Niño events (Fig. 1).”*

Ln 197: Here you refer to fig. 3 as “using the probability density function output from the statistical model...”, while fig. 3 shows climate and societal changes during the Holocene in MSEA and links with the end of the Green Sahara. In the first place, the reader cannot find immediately the information you are talking about, because you refer to the small part of the plot at the very top only at the very end of the caption. I suggest to move the density functions below in the figure, after subplot “f) Nile..”, naming it as “g) PDFs” or something like that, and in the text at Ln 197, next to Fig. 3, I would refer to Fig. 3g. Also please, make this part of the plot bigger...

Reply: Thank you, done.

Figure 4 and 6 in the text are not in vectorial format. Please check it.

Reply: Done.

“Discussion”. I think that this section about invoked mechanisms needs some further analysis, e.g. what is the relationship between ENSO and Walker circulation during the investigated season (JJAS)? ENSO peaks occur during NDJ, while you are investigating changes in Walker circulation during JJAS. How do you explain this inter-seasonal relationship?

Reply: We thank the reviewer for raising this important point. As such, we have attempted to improve the clarity of our discussion on the links between ENSO and the Walker circulation.

While ENSO does peak in December, it is most sensitive to perturbations applied in the spring/summer season (Thompson and Battisti, 2000). Therefore, the boreal summer is crucial for the development of the positive or negative ENSO phase. Studies focusing on modern climate have confirmed the link between Atlantic Niño, the Walker Circulation, and ENSO: Rodriguez-Fonseca *et al.* (2009) have shown that summer Atlantic Niño’s strengthen the Walker circulation with an intensified descending branch over the central Pacific, which favors La Niña conditions in the following winter as seen in our MH simulations (Pausata *et al.*, 2017). In a modeling study, Martin-Rey *et al.* (2012) have also shown that decreased SST variability in the equatorial Atlantic reduces ENSO activity. Indeed, using a coupled ocean-atmosphere model, these authors demonstrated that when tropical Atlantic SSTs are fixed and prescribed according to the observed monthly climatology, ENSO variability is lower than when the SSTs are interactively simulated in both

basins. In order to clarify the mechanism, we have accordingly modified the discussion, which now reads:

Line 307: *“Recent modeling experiments have demonstrated that the strengthening of the West African Monsoon (WAM) and the consequent Sahara “greening” played a dominant role in suppressing ENSO mean state and variability during the mid-Holocene⁵⁶, which might explain shifts in tropical hydroclimate through the drying of the Sahara at the end of the Green Sahara. While the ENSO phases peak in boreal winter, ENSO mode is most sensitive to perturbations applied in boreal summer from May through August⁵⁷. Specifically, Pausata et al. (ref. 56) demonstrated that a strengthened WAM led to warm SST anomalies and a reduction in SST variability over the equatorial Atlantic. This in turn caused the Walker circulation to shift westward, which can effectively influence ENSO activity and phases through changes in the strength of the trade winds in the equatorial Pacific. As shown in studies focusing on past⁵⁶ and modern climate^{58,59}, a westward shift of the Walker circulation causes an anomalous divergent flow during summer in the central-eastern Pacific, strengthening easterly winds over the western equatorial Pacific, while weakening them over the eastern side. The weaker trades reduce the upwelling and deepen the thermocline in summer over the eastern Pacific⁵⁶, reducing the atmosphere-ocean coupling and hence decreasing ENSO variability in agreement with several modeling studies (e.g., refs 60-62). On the other hand, the stronger trades in the central-western part of the basin causes a shoaling of the thermocline in the central Pacific during summer, leading to negative ocean temperature anomalies that travel eastward (Kelvin wave), reaching the eastern Pacific in boreal winter, and ultimately favoring the development of La Niña conditions^{56,58}.”*

LN: 248 - 257 This paragraph is not clear to me. It seems that weakened WAM is causing the ENSO reinforcement. Please also refer to "Towards understanding the suppressed ENSO activity during mid-Holocene in PMIP2 and PMIP3 simulations" by Lin Chen, Weipeng Zheng, Pascale Braconnot for further discussion.

Reply: We apologize for not being clear enough in explaining how the WAM failure impacted ENSO and the Walker circulation at the termination of the Green Sahara. In the revised version of the manuscript, we have rephrased that paragraph to better clarify the mechanism as shown in the previous response. We have also cited the Chen *et al.* (2019) manuscript in the discussion.

Reviewer #2 (Remarks to the Author):

There is one important point that the authors might consider, because it does reflect on how scholars in the other disciplines might interpret the results. What was the geographical extent of the drought area in Northern MSEA at about 5000 years ago? This is important, because the paper also raises, here and there, the likelihood that other regions to the immediate south, and in parts of China, actually received more rainfall at this time (e.g., lines 234–235 in the manuscript). In other words, the whole ‘megadrought’ concept does need very careful delineation in terms of exactly where the megadrought is considered to have happened. For cultural matters, such as migrations of human populations, this is likely to be very important.

Reply: We thank the reviewer for raising this important point, and completely agree that a nuanced discussion on the spatial extent of this megadrought across MSEA should be articulated in the paper. However, we feel that we have already provided such details so far as they are available in Figure 1 and Supplementary Table 2. Nonetheless, we have expanded the section “*Climate change and agrarian transitions in Southeast Asia*” and called out this pertinent regional climate evidence. We should also note that the archaeology of the most pertinent part of China, Yunnan, is still in rudimentary stages of investigation and bringing our climate data into the discussion will hopefully spur more sophisticated archaeological investigation in that region.

Reviewer #3 (Remarks to the Author):

The revised manuscript by Griffith et al. is an improvement of the previous version. The drought event occurred about 5.11~3.25 ka has been well supported by the stable isotope compositions, Mg/Ca ratio and the dead carbon proportions. By using the model simulations, it was proposed that reduced Saharan vegetation and increased dust emissions might contribute to the summer drought in northern Africa, the Arabian Peninsula, northern China, and the northern portions of both Indian and MSEA. This proposition is intriguing and deserved for studying by using both observational records and model simulations. However, there are still some issues need to be clarified before the manuscript could be published. I suggest a moderate reversion.

Reply: We thank the reviewer for the encouragement and overall positive and constructive reviews. The comments raised have greatly improved the quality of the manuscript.

Firstly, with the model simulations, I could accept that the end of green Saharan and increased dust emission may shift the Walker circulation eastward and cooled the Indian Ocean, causing a reduction in monsoon rainfall in MSEA and other regions. But I am curious that why the rainfall increased again while Saharan desert was still there and dust emissions was higher after 3.25 ka. The ending of this drought is also very important for us to understand the event. It suggests that a returning mechanism might be inherent to this event, e.g. recovering of the North Atlantic thermohaline circulations, although it is unlikely to be the primary driver as suggested by the authors. The authors should explain this accordingly, as it is closely related to the development of this event, and may provide clues to disclose the mechanism of this event.

Reply: We thank the reviewer for raising this important point and the subsequent recommendation. We are also intrigued by the sudden, but brief, return to ‘wet’ conditions following peak atmospheric dust loads. While we can only speculate at this stage, a number of potential scenarios may explain the hydroclimate ‘whiplash’ (i.e. abrupt transient dry, wet, dry shifts) that occurred between 4-2 ka. We have now added the following paragraph to the manuscript outlining these:

Line 262: *“It should be stressed, however, that factors other than Saharan dust and vegetation may have also influenced rainfall in MSEA during the mid-to-late Holocene. For example, the observed return to wetter conditions in MSEA between \approx 3-2.2 ka occurs despite the Sahara remaining generally dry. We note that similar patterns are seen in the Western Pacific SSTs⁵⁰ (Fig. 4b), whereby the mid-to-late Holocene decreasing trend was interrupted by a brief, but notable, increase at \sim 2.5 ka, suggesting a possible shift to La Niña-like conditions. It is also apparent in two of the presented East African δD_{wax} records^{3,47} (Figs 3b-c), further suggesting that this millennial scale return to wetter conditions was not restricted to MSEA. Furthermore, the dust records^{39,45,49} (Figs 3e-f) show that atmospheric dust loads were significantly higher between 4.2-3.5 ka compared with any other time during the Holocene, despite East Africa exhibiting a continued drying trend. While we can only speculate as to the driver(s) of this return to wetter conditions in MSEA around 3 ka, we hypothesize that the sudden abatement in dust loads after \approx 3.5 ka meant that other forcings and feedbacks (e.g., internal ENSO variability, Northern Hemisphere summer insolation, Atlantic Meridional Overturning Circulation) became more dominant.”*

Secondly, the authors state “The establishment and initial proliferation of this nucleated village agrarian life way in MSEA is remarkable in that that it occurred during a period of extreme climate variability across the broader East Asian region.” With the record reported in this study and other records mentioned, I agree to receive that the period from 5.1-3.2 ka is of extreme climate variability across the broader East Asia. However, there is no evidence or reference showing that the establishment and initial proliferation of this nucleated village agrarian life way occurred in MSEA during this period in the revised manuscript. The first appearance of cultivated cereals documented in MSEA, as mentioned in the text, is hardly to support this assessment. Although two broad periods, i.e., an early Holocene period and a late Holocene period, can be grabbed with the investigation of archeological records in this region, however, how and when the nucleated village agrarian life established and initially proliferated was still unclear. So I suggest the author to reorganize this section and demonstrate the link between the climate change and lifestyle change with more solid evidences. This is because the authors omitted the archeological evidence and discussions in the reversion.

Reply: *We thank the reviewer for this comment. In the revised version of the manuscript, we have revised this section and added some additional details linking the shift in human settlement patterns and megadrought in MSEA—see updated section “Climate change and agrarian transitions in Southeast Asia”.*

Here, the authors also refereed the linkage between weakened monsoon rainfall in the Yangtze River Basin and social and technological changes ca. 5.3-4.2 ka to indicate the climate change and its influences on human societies during the middle Holocene. However, as shown in the following sections, the drought within the middle Holocene in MSEA was largely linked with a dry climate in Northern China but a wet climate in central Eastern China. So the weakened monsoon rainfall in the Yangtze River Basin ca. 5.3-4.2 ka would be not the case. The authors need to double check

the fact and also the potential dynamics which may contradict with the fact.

Reply: We thank the reviewer for raising this point. We should note, however, that despite there being many published Holocene records from central China, there is still some ambiguity with respect to the regional homogeneity of rainfall variability during this period. For example, see figure below from a recent compilation of Holocene proxy records from East Asia showing rainfall anomalies during the ~4 ka period (Zhang *et al.*, 2018):

Figure 8. Map showing some locations discussed in the text. 1: Shennong Cave, this study; 2: Xiangshui Cave (Zhang *et al.*, 2004); 3: Heshang Cave (Hu *et al.*, 2008); 4: Sanbao Cave (Dong *et al.*, 2010); 5: Jiuxian Cave (Cai *et al.*, 2010); 6: Xianglong Cave (Tan *et al.*, 2018a); 7: Lianhua Cave (Dong *et al.*, 2015); 8: Nuanhe Cave (Tan, 2005); 9: Dongge Cave (Wang *et al.*, 2005); 10: Dark Cave (Jiang *et al.*, 2013); 11: Shigao Cave (Jiang *et al.*, 2012); 12: Xianren Cave (Zhang *et al.*, 2006); 13: Mawmluh Cave (Berkehammer *et al.*, 2012); 14: Erhai Lake (Zhou *et al.*, 2003); 15: Tianchi Lake (Zhao *et al.*, 2010); 16: Gonghai Lake (Chen *et al.*, 2015); 17: Daihai Lake (Xiao *et al.*, 2018a); 18: Dali Lake (Xiao *et al.*, 2008); 19: Hulun Lake (Xiao *et al.*, 2018b); 20: Daiyunshan peat (Zhao *et al.*, 2017); 21: Dahu peat (Zhou *et al.*, 2004); 22: Daping peat (Zhong *et al.*, 2010a); 23: Dajihu peat (Ma *et al.*, 2008); 24: Chengjiachuan site (Huang *et al.*, 2010); 25: Huxizhuang loess–soil profile (Huang *et al.*, 2011); 26: Tengchongqinghai Lake (Zhang *et al.*, 2017); 27: Gaochun profile (Yao *et al.*, 2017); 28: Zhongqiao site (Wu *et al.*, 2017). The solid triangle, dot and square denote stalagmite records, lake sediment–peat records and paleoflood sediment records, respectively. Black and blue indicate a dry and a wet climate during the 4.2 ka BP event, respectively. The base map is the same as that in Fig. 1.

From the above figure, it is apparent that while records from central eastern China show overwhelmingly wet/increased flood conditions at ~4 ka, there is greater uncertainty of the rainfall anomalies in the middle and upper reaches of the Yangtze River Basin; i.e., some records show drier conditions (black symbols) while others indicate wetter conditions (blue symbols) at around ~4 ka. Thus, in the interest of not excluding or favoring some records over others, we opted to present both views in the manuscript.

The open-system dissolution could well explain the decreasing trend of DCP started from beginning of the drought event. However, at the end of this event (e.g. 3-2 ka), the DCP data show a reversed relationship with the hydro-climate condition. How to interpret the data within this

interval is also very important for fully understanding this record.

Reply: Thank you. To address this comment, we have added the following to the manuscript:

Line 137: *“These results thus support a hydrologic interpretation of both $\delta^{13}\text{C}$ and Mg/Ca variation affected by PCP and local hydrology. The $\delta^{13}\text{C}$ and Mg/Ca records exhibit an abrupt return to values similar to 5.5 ka by ~ 3.3 ka, while the final DCP data points suggest recovery in this proxy may have been postponed up to $\sim 1,000$ years. This conflict could be a result of the disparate sampling resolutions between proxies and/or a delay in the response of the dissolution control on DCP vs. $\delta^{13}\text{C}$ and Mg/Ca controls.”*

Line 284-286, it was stated “ the westerlies were shifted southward between 4 and 5 ka (Fig. 6c), resulting in extreme drought conditions in Northern China and MSEA, and coincident with an equatorward contraction of the ITCZ.” Maybe I misunderstand here. The hypothesis for the drought event between 4-5 ka in MSEA is that the end of green Saharan and increased emission of dust lead to increased activity of ENSO and then delayed onset of Asian summer monsoon, not the southward shift of westerlies!

Reply: We thank the reviewer for this comment and we also apologize if we were not clear enough in our discussion of the links between ENSO, the ITCZ, and the westerlies.

As has been shown in the literature, modern El Niño events are generally characterized by a delay in the onset of the East Asian summer monsoon rains, due in large part to an equatorward contraction of the ITCZ (e.g., Berry and Reeder, 2014). While this equatorward contraction of the ITCZ can explain the lower latitude rainfall patterns during the mid-to-late Holocene (Fig. 6c, black and brown curves), it cannot, however, directly explain the extreme drying that occurred in northern China between 5-3.5 ka (Fig. 6c, red curve), because it is the westerlies which mostly heavily influence summer rainfall at these latitudes. Indeed, as noted in our manuscript, instrumental data show that the dominant mode of interannual rainfall variability across East Asia is the “tripole” mode, which is governed by the seasonal northward migration of the westerlies, and in particular, the position of these winds with respect to the Tibetan Plateau (Chiang *et al.*, 2017). Our model simulations demonstrate a significant delay and southward shift of the westerlies under the MH increased dust+reduced vegetation forcing (Fig. 6a), which can likely explain the extreme drying in northern China observed in our simulations (Figs 5a and 6b) and Lake Hulun record (Fig. 6c, red curve).

Critically, these observed rainfall patterns in the model are also consistent with the paleoclimate archives from central-eastern China, which overwhelmingly show wetter/increased flood conditions at ~ 4 ka (Fig. 1; also refer to Zhang *et al.*, 2018). The modelled shift in the westerlies is also supported by a paleoclimate record from the Japan Sea (Fig. 6c, blue curve; Nagashima *et al.*, 2013), which was interpreted to show a southward shift in the westerlies between 5-3.5 ka.

To summarize, our model simulations and paleoclimate archive comparisons, as shown in Fig. 6 and described in the manuscript, can most parsimoniously be explained by an interconnection between ENSO, the ITCZ, and the westerlies during this extreme climate event during the mid-to-late Holocene.

References

- Berry, G. and Reeder, M. J. Objective identification of the intertropical convergence zone: Climatology and trends from the ERA-Interim. *Journal of Climate* **27**, 1894-1909.
- Chen, L., Zheng, W., and Braconnot, P., 2019. Towards understanding the suppressed ENSO activity during mid-Holocene in PMIP2 and PMIP3 simulations. *Climate dynamics*, **53**, 1095-1110.
- Chiang, J., Swenson, L. and Kong, W., 2017. Role of seasonal transitions and the westerlies in the interannual variability of the East Asian summer monsoon precipitation. *Geophysical Research Letters* **44**, 3788-3795.
- Clement, A. C., Seager, R., and Cane, M. A., 1999. Orbital controls on the El Niño/Southern Oscillation and the tropical climate. *Paleoceanography*, **14**, 441-456.
- Liu, Z., Kutzbach, J., and Wu, L., 2000. Modeling climate shift of El Niño variability in the Holocene. *Geophysical Research Letters* **27**, 2265-2268.
- Martín-Rey, M., Polo, I., Rodríguez-Fonseca, B., and Kucharski, F., 2012. Changes in the interannual variability of the tropical Pacific as a response to an equatorial Atlantic forcing. *Scientia Marina* **76**, 105-116.
- Pausata, F. S., Zhang, Q., Muschitiello, F., Lu, Z., Chafik, L., Niedermeyer, E. M., and Liu, Z., 2017. Greening of the Sahara suppressed ENSO activity during the mid-Holocene. *Nature communications* **8**, 1-12.
- Rodríguez-Fonseca, B., Polo, I., García-Serrano, J., Losada, T., Mohino, E., Mechoso, C. R., and Kucharski, F., 2009. Are Atlantic Niños enhancing Pacific ENSO events in recent decades? *Geophysical Research Letters*, **36** L20705.
- Thompson, C. J., and Battisti, D. S., 2000. A linear stochastic dynamical model of ENSO. Part I: Model development. *Journal of Climate* **13**, 2818-2832.
- Zhang, H., Cheng, H., Cai, Y., Spötl, C., Kathayat, G., Sinha, A., and Tan, L., 2018. Hydroclimatic variations in southeastern China during the 4.2 ka event reflected by stalagmite records. *Climate of the Past*, **14** (11).

Reviewers' comments second round:

Reviewer #1 (Remarks to the Author):

Dear Authors,

I found pretty exhaustive the point-by-point replies. Thanks for the nice work. I think that the manuscript is overall improved although I did not find any cross-analysis to support the objections raised by reviewers. In my view, it is very complicated to address causality relationship with time-slice simulations and hence, I suggest the editor to ask a third reviewer a feedback on the data&methods, specifically a third-opinion on the simulations used in this study would be beneficial.

Reviewer #3 (Remarks to the Author):

The revised manuscript responds to most of the comments and suggestions by reviewers. I can make no substantive suggestions for improvement. I think that the manuscript is now suitable for publication in Nature Communications.

End of Green Sahara amplified Mid-to-Late Holocene megadroughts in the Middle Mekong Basin

Michael L. Griffiths, Kathleen R. Johnson, Francesco S.R. Pausata, Joyce C. White, Gideon M. Henderson, Christopher T. Wood, Hongying Yang, Vasile Ersek, Cyler Conrad, Natasha Sekhon

Reviewer #1 (Remarks to the Author):

I found pretty exhaustive the point-by-point replies. Thanks for the nice work. I think that the manuscript is overall improved although I did not find any cross-analysis to support the objections raised by reviewers. In my view, it is very complicated to address causality relationship with time-slice simulations and hence, I suggest the editor to ask a third reviewer a feedback on the data&methods, specifically a third-opinion on the simulations used in this study would be beneficial.

Reply: We thank the referee for their support of this work. We feel that we addressed all of the reviewer's comments in the previous submission, and provided sufficient rebuttals where appropriate.

Reviewer #2 (Remarks to the Author):

“I am not a paleoclimatologist, so I do not comment on the climatological aspects of this paper. The suggestion that the northern regions of Mainland Southeast Asia, centered on Northern Laos and Northeast Thailand, experienced a major drought about 4000 years ago seems to me to be quite in accord with general views about other parts of the world, for instance China and the Sahara.

My main focus is on the archaeological section, which I understand to have been rewritten. The current section on archaeology makes the suggestion that major climate change has not heretofore been considered as a driving factor in the societal shifts that occurred during the mid-Holocene in MSEA. This is true, but I challenge anyone to prove that a demonstrated climate change was a ‘cause’ of a change in culture, even if both were contemporary (as they presumably were, in this instance). Climate and people are like apples and oranges – they do not grow on the same tree.

As the paper stands (third time around I think), I noticed other puzzling matters:

- The archaeological text refers only to sites in Thailand. But other very similar Neolithic sites, dating from around 2000 BC and onwards, are found all over the mainland of SE Asia, including Vietnam, Burma, Malaysia, plus Taiwan, Philippines, etc, not to mention southern China. There are hundreds of them. If all of these regions underwent the drought, fine, but I do not see any claim for this, and it is not shown thus in Fig. 1.

- On the other hand, there is a large quantity of archeological, ancient DNA, modern DNA, craniometric, zooarchaeological, archeobotanical, and linguistic evidence that shows that the observed archaeological changes were due to movements of agricultural populations from further north. I can understand that a paper such as this cannot possibly consider such a huge multidisciplinary literature. But it does seem to be rather ambivalent about immigration and it

seems to want it swept under the carpet as much as possible, even if it can't quite get rid of it altogether.

- It also seems to me that an unexpected drought might drive people out of a given area, and the authors note the Yangzi Basin was apparently depopulated when drought hit about 4000 years ago. But we would have to wonder why migrants would head for another area that was also suffering from drought. But, then again, a migration of existing agricultural populations from an exterior location would probably have been unaffected by local drought conditions, because people would in effect have been starting again with respect to land distribution. They would have sought places close to water, if they existed. And if they did, climate change might have been irrelevant as long as all the water sources had not dried up completely.

In conclusion, I do not disagree with the statement that "...population movements may have been instigated in part by mid-to-late Holocene climate changes and those movements in turn may have introduced cereal agriculture into MSEA." But I do not think that the evidence presented in this paper necessarily demonstrates this to be true. One might never be able to demonstrate it. As far as the archeological section of this paper is concerned, I would prefer to see a simple statement that the climate change occurred around the same time that agricultural populations migrated from China into Mainland Southeast Asia. Such a statement could be linked to the simple question – was it, or was it not, coincidence?"

Reply: The referee raises some excellent points regarding the limitations of our extended discussion on the potential links between societal movements/adaptations and the megadroughts in MSEA. In light of this, we have significantly "toned down" this section of the manuscript, and thus reverted it back to our previous submission, with the exception of adding a final sentence in response to the reviewer's suggestion. To this section, we have also added a very recent and pertinent reference connecting cereal agricultural spread to climate change in East Asia during the Mid-to-Late Holocene—Gutaker *et al.*, 2020, *Nature Plants* (citation #32 in our manuscript).

Reviewer #3 (Remarks to the Author):

The revised manuscript responds to most of the comments and suggestions by reviewers. I can make no substantive suggestions for improvement. I think that the manuscript is now suitable for publication in Nature Communications.

Reply: We thank the referee for the encouragement and support of our manuscript.

Report from reviewer #4

"Review's comments for revised manuscript NCOMMS-19-41506-A, entitled "Mid-to-Late Holocene Megadroughts in the Middle Mekong Basin Linked to Global Climate Changes", submitted to Nature Communications

General comments

By exploiting geological evidences, this paper identify a megadrought event during the middle Holocene in Mainland South East Asian (MSEA) during summer season. The study further argues

that the reduced vegetation and increased dust loads during the end of the Green Sahara are likely the main cause. The mechanisms involved are changes in the Walker circulation and resulted changes in tropical Pacific mean state and ENSO variability. The study is novel and results are interesting. The reviewer did not referee previous versions of this paper. By reading authors' responses and revised paper, the reviewer thinks authors have addressed reviewers' comments on an early version of this paper in a satisfactorily manner. The revised paper is well written and it is acceptable for publication."

Reply: We thank the referee for their support of our manuscript for publication.